# Serum Phosphate Levels Modify the Impact of FGF23 Levels on Hemoglobin in Chronic Kidney Disease

**DOI:** 10.3390/nu14224842

**Published:** 2022-11-16

**Authors:** Juan F. Navarro-González, Carmen Mora-Fernández, Juan Miguel Diaz-Tocados, Milica Bozic, Marcelino Bermúdez-López, Marisa Martín, Jose Manuel Valdivielso

**Affiliations:** 1Research Unit, University Hospital Nuestra Señora de Candelaria, 38010 Santa Cruz de Tenerife, Spain; 2Nephrology Service, University Hospital Nuestra Señora de Candelaria, 38010 Santa Cruz de Tenerife, Spain; 3Vascular and Renal Translational Research Group, Biomedical Research Institute of Lleida, Dr. Pifarré Foundation (IRBLleida), Rovira Roure 80, 25198 Lleida, Spain; 4Department of Medicine, University of Lleida, 25003 Lleida, Spain

**Keywords:** chronic kidney disease, hemoglobin, FGF23, serum phosphate, anemia

## Abstract

Anemia is a complication of chronic kidney disease (CKD). Phosphate and fibroblast growth factor-23 (FGF23) have a close relationship, as both are related to the pathogenesis of anemia. However, the possible interplay between them regarding their effect on anemia has not been evaluated. This was a cross-sectional study of 896 participants from the NEFRONA study (273 CKD3, 246 CKD4-5, 282 dialysis and 95 controls). The levels of 25(OH) and 1,25(OH)_2_ vitamin D, intact FGF23 (iFGF23) and soluble Klotho were measured, together with standard blood biochemistries. Anemia was defined as hemoglobin levels < 13 g/dL in men and <12 g/dL in women. Patients with anemia (407, 45.4%) were younger, mostly men and diabetic; were in advanced CKD stages; had lower calcium, 1,25(OH)_2_ vitamin D and albumin levels; and had higher ferritin, phosphate, intact PTH, and iFGF23. An inverse correlation was observed between hemoglobin and both iFGF23 and phosphate. The multivariate logistic regression analyses showed that the adjusted risk of anemia was independently associated with higher serum phosphate and LogiFGF23 levels (ORs (95% CIs) of 4.33 (2.11–8.90) and 8.75 (3.17–24.2), respectively (*p* < 0.001)). A significant interaction between phosphate and iFGF23 (OR of 0.66 (0.53–0.83), *p* < 0.001) showed that the rise in the adjusted predicted risk of anemia with the increase in iFGF23 was steeper when phosphate levels were low. Phosphate levels acted as modifiers of the effect of iFGF23 concentration on anemia. Thus, the effect of the increase in iFGF23 levels was stronger when phosphate levels were low.

## 1. Introduction

Anemia is a common complication of chronic kidney disease (CKD) whose prevalence and severity increases as renal function declines. This complication is associated with a myriad of deleterious consequences in the context of CKD, including reduced quality of life, increased risk of cardiovascular morbidity (development of left ventricular hypertrophy, angina and congestive heart failure), adverse cardiovascular outcomes and mortality and higher costs, as well as raised risk of renal function deterioration and progression of CKD [1,2,3]. The kidney is the main source of erythropoietin (EPO) [4], and progressive destruction of renal parenchyma decreases its availability, with this being one of the main causes of CKD-related anemia [5]. However, EPO deficiency is not the only cause. Indeed, iron loss and the excess of hepcidin have also been held responsible for this complication [6]. Furthermore, numerous studies suggest that circulating inhibitors of erythropoiesis contribute to anemia in the uremic state [6].

Hyperphosphatemia and elevations of fibroblast growth factor 23 (FGF23) are common findings in CKD, with relevant pathophysiological roles regarding diverse CKD complications, including anemia. Thus, several clinical studies have reported the association between hyperphosphatemia and anemia in different CKD settings, including dialysis and transplantation [7,8,9], as well as the role of hyperphosphatemia as a risk factor for hyporesponsiveness to erythropoiesis-stimulating agents (ESAs) [7,8]. Regarding FGF23, experimental studies have demonstrated that the administration of this factor leads to a reduction in erythropoiesis, while the loss of the effect of this molecule results in opposite changes, with a significant increase in red blood cell production [10,11]. However, these experimental findings have not been fully corroborated in clinical studies, with some of them showing that FGF23 concentrations are associated with reduced hemoglobin levels [12,13,14], while others show no associations [15,16]. Since serum FGF23 levels rise early in the course of CKD, elevated FGF23 may represent a novel contributor to the development of anemia in CKD.

There is a close interplay between FGF23 and phosphate (P), since FGF23 is regulated by feedback loops involving the P status [17]. However, it is not yet clear if the effects of FGF23 and P are independent and whether the levels of one can modify the effects of the other on anemia. Furthermore, some treatments for anemia in CKD have potential effects on FGF23 and P concentrations, which can have an impact on the effectiveness of these therapies, raising hemoglobin levels [18]. Therefore, a deeper understanding of the relationship between these parameters and anemia is of paramount importance.

In the present study, we evaluated the potential interaction between P and FGF23 on hemoglobin levels and renal anemia in a relatively large cohort of patients in different CKD stages.

## 2. Materials and Methods

### 2.1. Study Population

This was a cross-sectional study of a subcohort from the NEFRONA study that included 801 CKD patients (273 in CKD stage 3, 246 in CKD stages 4–5 and 282 on dialysis) and 95 controls for which fasting serum samples were available. NEFRONA (National Observatory of Atherosclerosis in Nephrology) is a prospective multicenter cohort study in which 2445 CKD subjects were enrolled in 81 Spanish hospitals and dialysis clinics, from October 2010 to June 2012. Patients between 18 and 74 years of age were eligible if they had CKD stage 3 or higher as defined by current guidelines (glomerular filtration rate (GFR) lower than 60 mL/min/1.73 m^2^, estimated using the 4-variable Modification of Diet in Renal Disease (MDRD) equation). Additionally, 559 controls with an MDRD value over 60 mL/min/1.73 m^2^ were recruited from primary care centers, matched by sex and age. The NEFRONA detailed methods have already been published elsewhere [19,20]. Recruiting investigators completed a questionnaire with the patients’ clinical data, including family history of early cardiovascular disease, cardiovascular risk factors (such as smoking, diabetes, hypertension or dyslipidaemia) and current medications. Anthropometrical data and vitals (height, weight, waist circumference, blood pressure) were obtained using standardized methods, as described in the rationale publication. The protocol of the study was approved by the ethics committee of each participating hospital, and all patients signed their informed consent. The study was conducted according to the principles of the Declaration of Helsinki.

### 2.2. Clinical Data and Laboratory Determinations

The GFR was estimated according to international guidelines using the MDRD equation. CKD patients were classified regarding their GFR into 3 groups: CKD stage 3, CKD stages 4–5 and patients on dialysis. The diagnoses of dyslipidemia, diabetes and hypertension were obtained from the clinical history. The diagnosis of anemia was assigned if the levels of hemoglobin were below 13 g/dL in men or below 12 g/dL in women. The rest of the anthropometrical parameters and smoking habits were obtained at the moment of collection of the fasting blood samples. After centrifugation, serum samples were aliquoted to avoid freeze–thaw cycles and stored at −80 °C until the determination of high-sensitivity C reactive protein (hsCRP), soluble KLOTHO (sKLOTHO), 1,25 and 25OH Vit D and intact FGF23 (iFGF23) was performed. The biochemical parameters were acquired from a routine fasting blood test performed no more than three months after the inclusion in the study. In hemodialysis patients, samples were obtained before the second dialysis session of the week.

Since NEFRONA is a multicenter study, particular attention was paid to those parameters measured with different methods. In particular, intact PTH (iPTH) for the dialysis group was corrected using a well-established conversion method [21]. For patients without dialysis treatment, the correction was not applied, because residual or normal renal function decreases the formation of the 7–84 PTH fragments, which, at high levels, are the responsible for high inter-method differences. Furthermore, the levels of hsCRP, 1,25 and 25-OH vitamin D were measured in the same laboratory to avoid possible interferences due to the use of different methods. hsCRP was determined with an immunoturbidimetric method (Roche/Hitachi MODULAR ANALYTICS, Barcelona, Spain), and 1,25 and 25-OH vitamin D levels were determined with an ELISA (IDS, Brighton, UK).

iFGF23 levels were assayed with an Immutopics Human ELISA kit (Immutopics Inc., San Clemente, CA, USA), which presents a sensitivity of 1.5 relative units (RU)/mL and intra- and inter-assay coefficients of 1.9% and 3.55%, respectively. sKLOTHO was measured with a solid-phase sandwich ELISA using a human soluble α-Klotho assay kit (Immuno-Biological Laboratories, Takasaki, Japan), with a sensitivity of 6.15 pg/mL and intra- and inter-assay coefficients of variation <3.1% and 6.9%, respectively.

### 2.3. Statistical Analysis

Descriptive analyses included the absolute and relative frequencies for qualitative variables, as well as the median and quartiles 1 and 3 for quantitative variables. The Pearson chi-square test was used to compare the distribution of qualitative variables between anemic and non-anemic patients. The non-parametric Mann–Whitney test was used to compare the distribution of quantitative variables between anemic and non-anemic patients. The independent association between P and iFGF23, and the presence of anemia was achieved by fitting a multivariate logistic regression model. The highly skewed levels of iFGF23 were converted using a logarithmic transformation. The interaction effects between P and iFGF23 levels were tested and included in the model only if statistically significant. Several multivariate models were tested. First, crude ORs were calculated for P, iFGF23 and the interaction of both factors. A second model was adjusted for sex and age, and a third model was further adjusted for all variables found to be significantly different in the bivariate analysis, plus any potential confounders. The statistical level of significance was fixed at 0.05. The SPSS program was used for all statistical analyses.

## 3. Results

### 3.1. Description of the Study Cohort

The clinical characteristics and bivariate analyses of the patients are shown in Table 1. Patients presenting with anemia were mostly men, younger, diabetic and in more advanced stages of CKD. Anemic patients showed a higher use of ESAs and intravenous (IV) iron therapy. As expected, hemoglobin levels were lower in subjects with anemia, but ferritin levels were higher. Furthermore, patients with anemia showed significantly lower serum calcium, albumin and 1,25(OH)2D3 concentrations and higher serum P, iPTH and iFGF23.

### 3.2. Relationships of Serum P and FGF23 with Anemia

As serum P and FGF23 concentrations potentially contribute to the risk of anemia, we studied the association between the hemoglobin levels and both P and iFGF23 levels. The correlation analysis showed an inverse and significant association between hemoglobin concentration and both serum iFGF23 and P (Figure 1).

To further compare the relative prevalence of iFGF23 versus P levels with the presence of anemia, we studied the association between hemoglobin concentrations and low (below the median) or high (over the median) iFGF23 and P levels in women and men (iFGF23 median, 137.6 RU/mL; P median, 3.8 mg/dL). We observed that patients within the low-iFGF23 and low-P groups showed higher hemoglobin concentrations, whereas those within the high-iFGF23 and/or high-P groups showed decreased hemoglobin levels (Figure 2). In every case, women showed lower hemoglobin concentrations than men.

### 3.3. Multivariate Analysis

Exploratory multivariate linear regression analyses were used in order to determine the possible association between iFGF23 and P, and hemoglobin levels. In order to simplify the interpretation, a multivariate logistic regression analysis of factors associated with anemia was performed (Table 2). The results show that both iFGF23 and P, together with their interaction, were independently associated with a higher risk of anemia in the crude analysis (Model 1), after adjustment for sex and age (Model 2) and in the fully adjusted model (Model 3, further adjusted for diabetes, CKD stage, albumin, ferritin, calcium, iPTH, hsPCR, 25(OH) vitamin D, 1,25(OH)2 vitamin D, IV iron therapy and ESA treatment).

The effect of the interaction between serum P and iFGF23 is depicted in Figure 3. As shown, the adjusted predicted probabilities of anemia increased with the increase in iFGF23. However, the effect was different depending on the P levels, as the slope in the low-P group was higher than that in the high-P group.

## 4. Discussion

In the present study, we investigated the relationship between iFGF23 and anemia, as well as the possible effect of P modifying the association. The results clearly show that although both compounds are inversely associated with hemoglobin levels, P acts as an effect modifier of the iFGF23 association with anemia, after extensive adjustment for many other potential confounding risk factors. Thus, the effect of increased iFGF23 levels on anemia is stronger when P levels are low.

Anemia is a serious complication in CKD patients, in which elevations of FGF23 levels are an early feature. Although defects in EPO and iron handling have been described as important features of CKD involved in anemia development, several other factors have also been suggested. Thus, chronic inflammation, malnutrition, increased destruction of red blood cells and vitamin D deficiency also contribute to the pathogenesis of renal anemia [22,23,24]. In addition, the involvement of FGF23 levels in anemia in CKD has also been suggested. However, studies evaluating this relationship have yielded conflicting results. In a study by Akalin et al., the authors showed a lack of association between FGF23 and hemoglobin levels in 89 hemodialysis patients [15]. Likewise, a study by Honda et al. in a larger hemodialysis cohort showed no significant relationship between FGF23 and hemoglobin levels [16]. However, those findings were contested by other studies in patients undergoing peritoneal dialysis or patients with CKD before dialysis [12,13,14,25] in which higher levels of FGF23 were associated with higher odds of anemia.

Very recent results from Czaya et al. showed an independent effect of elevated P levels on anemia [26]. The effects seemed to be independent of FGF23 signaling, at least through its FGF4 receptor, as the effect was maintained in FGF4 null mice. Previous clinical studies also found an association between hyperphosphatemia and lower hemoglobin concentrations [7,8,9]. However, whether those effects were maintained when the levels of FGF23 were included in the analyses is unknown. Furthermore, the well-known effect of intravenous iron treatments on P levels adds a layer of complexity in analyzing the association and calls for further adjustment factors in the statistical analyses [18]. Thus, our results could explain some of the inconsistent findings in previous studies assessing the association between FGF23 and anemia. In the studies of Akalin and Honda [15,16], higher levels of P could have masked the true association between FGF23 and anemia. Indeed, those studies were performed in hemodialysis patients, a population in which P levels are usually higher than in other CKD stages [27].

How P levels can modify the effect of iFGF23 in anemia in CKD is unknown. However, we could hypothesize that with the same levels of iFGF23, higher levels of P would indicate resistance of the kidney to the actions of the hormone, reflecting the decrease in renal Klotho that has been described in CKD [28]. Indeed, it has been suggested that CKD is a state of systemic Klotho deficiency [29]. The expression of Klotho has been detected in bone marrow cells both in rodents [30] and humans [31], and FGF23 from erythroblasts has been shown to promote hematopoietic stem cell mobilization [32]. Thus, studies in Klotho KO mice have revealed that a lack of klotho adversely affects hematopoiesis [33]. Therefore, the results showing a decreased effect of the increase in the concentrations of iFGF23 when P levels are high could point to a decrease in Klotho in erythropoietic cells in CKD.

Our study had some limitations. First, its cross-sectional nature precluded us from obtaining any causal relationship. Second, we only measured iFGF23 levels. Therefore, the contribution of other fragments of FGF23 could not be assessed. However, the analysis also showed some strengths. Thus, the relative high number of patients allowed us to adjust the model for many variables that could be confounding for the outcome. Furthermore, the centralization of the determinations of several parameters (including iFGF23) avoided a possible operator bias.

## 5. Conclusions

In conclusion, the effect of serum iFGF23 levels on anemia is modified by serum P concentrations. A close monitoring of both parameters could help to maintain hemoglobin levels in range in CKD patients.

## Figures and Tables

**Figure 1 nutrients-14-04842-f001:**
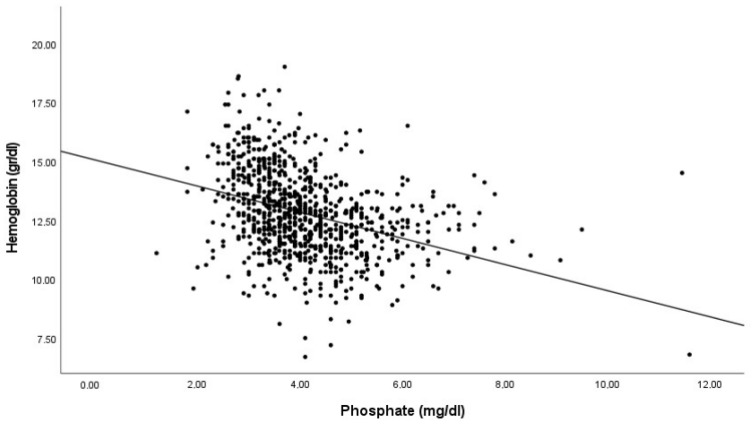
Correlation between hemoglobin levels with FGF23 and phosphate. Pearson’s correlation coefficients −0.352 and −0.423 for Log FGF23 and Phosphate respectively. *p* < 0.001 in both cases.

**Figure 2 nutrients-14-04842-f002:**
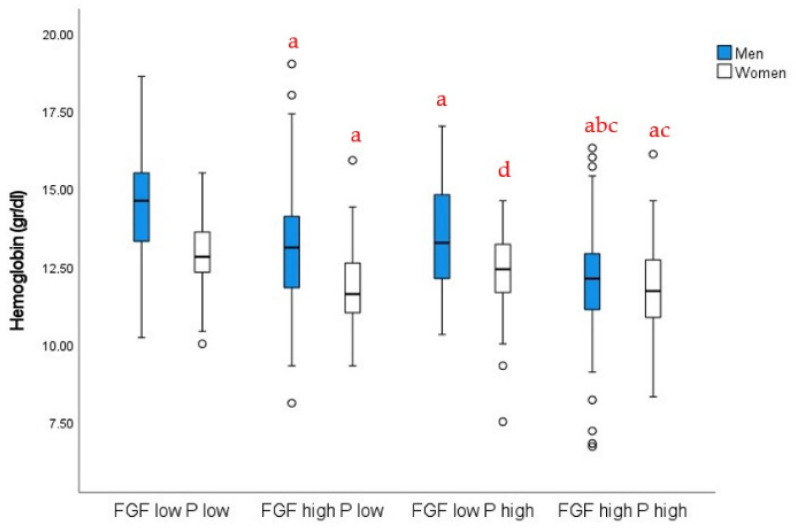
Hemoglobin levels in men and women depending on P and iFGF23 levels: a, *p* < 0.01 vs. low FGF–low P; b, *p* < 0.01 vs. high FGF–low P; c, *p* < 0.01 vs. low FGF–high P; d, *p* < 0.05 vs. low FGF–low P, same sex. Two-way ANOVA with post hoc Bonferroni test. Interaction *p* < 0.01.

**Figure 3 nutrients-14-04842-f003:**
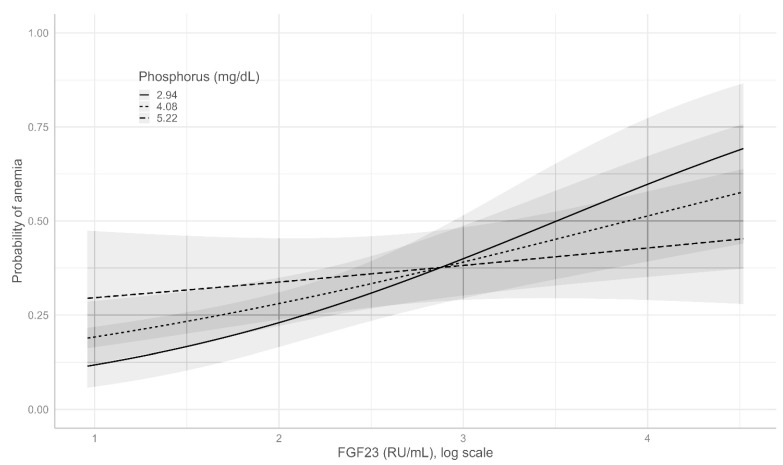
Adjusted predicted probability (95% CI) of anemia with different values of iGF23 and serum phosphate.

**Table 1 nutrients-14-04842-t001:** Clinical and biochemical characteristics of the total subcohort and bivariate analysis of differences between patients with and without anemia.

	ALL	ANEMIA		*p*
		NO	YES	
*n* (%)	896 (100)	489 (54.6)	407 (45.4)	
Age (years)	59 (48; 68)	62 (51; 68)	57.0 (44; 67)	**0.003**
Sex (women)	361 (40.3)	212 (43.4)	149 (36.6)	**0.04**
Diabetes	178 (19.9)	80 (16.4)	98 (24.1)	**0.004**
CKD stage				
Control	95 (10.6)	89 (18.2)	6 (1.5)	Ref.
CKD3	273 (30.5)	217 (44.4)	56 (13.8)	**0.012**
CKD4–5	246 (27.5)	102 (20.9)	144 (35.4)	**<0.001**
Dialysis	282 (31.5)	81 (16.6)	201 (49.4)	**<0.001**
ESA use	318 (35.5)	84 (17.2)	234 (57.5)	**<0.001**
IV iron therapy	407 (45.4)	55 (11.2)	122 (30.0)	**<0.001**
BMI (kg/m^2^)	27.4 (24.3; 30.9)	28.1 (25.0; 31.2)	26.7 (23.7; 30.4)	0.092
SBP (mmHg)	143 (129; 157)	141 (129; 153)	145 (128; 161)	0.067
DBP (mmHg)	82.0 (75.0; 90.0)	83.0 (77.0; 89.0)	81.0 (73.0; 90.0)	0.243
Hemoglobin (g/dL)	12.6 (11.5; 13.7)	13.7 (13.0; 14.9)	11.5 (10.9; 12.1)	**<0.001**
Ferritin (ng/mL)	170 (81; 322)	135 (71; 263)	214 (102; 412)	**<0.001**
Calcium (mg/dL)	9.3 (8.9; 9.6)	9.4(9.0; 9.7)	9.1 (8.8; 9.5)	**<0.001**
Phosphate (mg/dL)	3.9 (3.3; 4.6)	3.6 (3.1; 4.2)	4.3 (3.7; 5.0)	**<0.001**
Albumin (g/dL)	4.1 (3.8; 4.4)	4.2 (3.8; 4.5)	4.0 (3.8; 4.4)	**0.002**
iPTH (pg/mL)	137 (74; 237)	98 (59; 170)	184 (101; 276)	**<0.001**
sKlotho (pg/mL)	288 (176; 437)	289 (169; 447)	282 (183; 416)	0.434
Log iFGF23 (pg/mL)	2.29 (1.90; 2.86)	2.06 (1.76; 2.47)	2.56 (2.12; 3.09)	**<0.001**
HsCRP(mg/L)	1.98 (1.00; 4.62)	1.83 (0.94; 4.24)	2.44 (1.17; 6.55)	0.145
25-OH D (ng/mL)	15.2 (11.5; 19.8)	15.0 (11.2; 19.7)	15.4 (11.8; 19.9)	0.243
1,25-OH_2_ D (pg/mL)	13.8 (8.4; 21.6)	15.9 (9.8; 25.2)	11.7 (7.3; 18.4)	**<0.001**

Data are presented as numbers (percentages within the anemia group) for qualitative variables and medians (Q1; Q3) for quantitative variables. *p*-values represent chi-square or *t*-tests for qualitative and quantitative variables, respectively. ESA, erythropoietin stimulating agents; IV, intravenous; BMI, body mass index; SBP, systolic blood pressure; DBP, diastolic blood pressure; iPTH, intact parthyropid hormone; HsCRP, high-sensitivity C-reactive protein.

**Table 2 nutrients-14-04842-t002:** Multivariate logistic regression analysis results of the association between iFGF23, phosphate and their interaction, and the presence of anemia.

	Model 1		Model 2		Model 3	
Variable	OR (95% CI)	*p*	OR (95% CI)	*p*	OR (95% CI)	*p*
Phosphate (mg/dL)	7.9 (4.15–14.9)	**<0.001**	9.5 (4.93–18.3)	**<0.001**	4.33 (2.11–8.90)	**<0.001**
Log iFGF23 (pg/mL)	21.1 (8.43–52.9)	**<0.001**	26.6 (10.4–67.9)	**<0.001**	8.75 (3.17–24.2)	**<0.001**
Phosphorus by Log iFGF23	0.57 (0.47–0.70)	**<0.001**	0.54 (0.44–0.66)	**<0.001**	0.66 (0.53–0.83)	**<0.001**

**Model 1:** Crude analysis. **Model 2:** adjusted for age and sex. **Model 3:** adjusted for age, sex, diabetes, CKD stage, albumin, ferritin, calcium, intact PTH, usPCR, 25(OH) vitamin D, 1,25(OH)_2_ vitamin D, intravenous iron therapy and erythropoietin stimulating agents treatment. AUC (95% CI) ROC curve: 0.809 (0.776, 0.841). Hosmer Lemeshow *p*: 0.404.

## Data Availability

Data are available upon request from the corresponding author.

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
