# Peer review of "Serum Phosphate Levels Modify the Impact of FGF23 Levels on Hemoglobin in Chronic Kidney Disease"

_nutrients, 2022, doi:10.3390/nu14224842_

Round 1
Reviewer 1 Report
In this work, the authors investigated the relationship between FGF23 and anemia and the possible effect of P modifying the association. The research work is meaningful and the writing language of this manuscript is also standardized. Therefore, the manuscript can be accepted directly when a few minor problemes were revised.
In previous studies, alterations in mineral metabolism disorders with progression of CKD increased PTH before abnormal serum phosphorus and Ca concentrations occurred. The author describes that patients with anemia showed significantly lower serum calcium, albumin and 1,25(OH)2D3 concentrations and higher serum P, PTH and iFGF23.
What do you think about serum Ca, ferritin levels, and PTH were correlated?
Author Response
In this work, the authors investigated the relationship between FGF23 and anemia and the possible effect of P modifying the association. The research work is meaningful and the writing language of this manuscript is also standardized. Therefore, the manuscript can be accepted directly when a few minor problemes were revised.
In previous studies, alterations in mineral metabolism disorders with progression of CKD increased PTH before abnormal serum phosphorus and Ca concentrations occurred. The author describes that patients with anemia showed significantly lower serum calcium, albumin and 1,25(OH)2D3 concentrations and higher serum P, PTH and iFGF23.
What do you think about serum Ca, ferritin levels, and PTH were correlated?
We thank the reviewer for the suggestions. We have explored correlations between Ca, PTH and ferritin in patients with and without anemia. Calcium showed a significant negative correlation with both PTH and ferritin in dependently of the presence of anemia and with very similar correlation coefficients. However, the significant positive correlation of ferritin with PTH disappeared in anemic patients. The explanation for this fact is unclear and complicated, but we could hypothesize that the stimulus for anemia in these patients will have a strong effect increasing ferritin levels but not PTH. However, that same stimulus will decrease Ca levels with a similar strength as the correlation Ca-ferritin is maintained with similar coefficient in anemia. The nature of that stimulus is beyond the scope of this study.
|
|
NO anemia |
Anemia |
||||
|
Ca |
PTH |
Ferritin |
Ca |
PTH |
Ferritin |
|
|
Ca |
|
-0.329*** |
-0.127** |
|
-0.266*** |
-0.116* |
|
PTH |
|
|
0.211*** |
|
|
0.073 |
***: p<0.001; **: p=0,005; *: p=0.019
Reviewer 2
Navarro-Gonzalez and colleagues investigated the role of P, FGF23, 25(OH), 1,25(OHD)2, PTH on anemia in CKD.
The research question is of interest and the methods well performed.
Below some comments:
1) Introduction. It would be good to have in the introduction more background on the effect that anemia has on CKD.
We thank the reviewer for the suggestion. We have now expanded the introduction with more data regarding the impact of anemia in CKD.
2)It would be good to have FGF23 and PTH consistently named throughout the text.
We have checked for accuracy and consistency in the denomination of FGF23 and PTH throughout the text. We have denominated iFGF23 to the measures we have performed in our study (together with iPTH). However, when referring to other studies, we have maintained FGF23 or PTH as the exact length of the protein measured is missing.
3) Figure 2 could be integrated with p-value. The cut-offs value for low FGF23 and Low P could be specified in the legend.
We apologize for not including the p values in figure 2. The figure now includes those values and the text immediately above the median values of FGF23 and P used for categorizing the variables.

Reviewer 2 Report
Navarro-Gonzalez and colleagues investigated the role of P, FGF23, 25(OH), 1,25(OHD)2, PTH on anemia in CKD.
The research question is of interest and the methods well performed.
Below some comments:
1) Introduction. It would be good to have in the introduction more background on the effect that anemia has on CKD.
2)It would be good to have FGF23 and PTH consistently named throughout the text.
3) Figure 2 could be integrated with p-value. The cut-offs value for low FGF23 and Low P could be specified in the legend.
Author Response

(The authors gave the same response as above.)
